# A Pilot Study to Assess the Feasibility of Real-Time Teledentistry in Residential Aged Care Facilities

**DOI:** 10.3390/healthcare12222216

**Published:** 2024-11-06

**Authors:** Cheuk Kee Candy Fung, Diep Hong Ha, Laurence James Walsh, Claudia Patricia Lopez Silva

**Affiliations:** 1School of Dentistry, The University of Queensland, Brisbane 4006, Australial.walsh@uq.edu.au (L.J.W.); c.lopezsilva@uq.edu.au (C.P.L.S.); 2Oral Health Centre, Metro North Oral Health Services, Queensland Health, Brisbane 4006, Australia

**Keywords:** teledentistry, real-time videoconferencing, oral healthcare, oral hygiene, residential aged care facilities, nursing, cost analysis

## Abstract

Background/Objectives: Unmet oral health needs of residents in residential aged care facilities (RACFs) arise due to the unique challenges of assessing oral health statuses and maintaining oral healthcare in RACFs. This pilot study assessed the feasibility of using real-time teledentistry under the guidance of a dentist to train RACF staff to undertake an oral health assessment. Methods: An oral health assessment of residents was first conducted by RACF staff at two Queensland, Australia RACFs using the Oral Health Assessment Tool, with an intra-oral camera connected to a laptop, through videoconferencing, under the guidance of a dentist. A survey recorded the views of RACF staff on the acceptability of the teledentistry method. The quality of the images obtained through the camera was assessed by the dentist. Finally, cost-effectiveness was calculated between teledentistry and traditional face-to-face assessments. Results: Sixteen residents (mean age 79.3 ± 8.68 years) and eight staff (mean age 33.3 ± 6.16 years) participated in this study. Both RACF staff and residents found that the real-time teledentistry set-up was user-friendly, while the dentist rated the quality of the images as acceptable for diagnostic purposes. Real-time teledentistry was more cost-effective than bringing a dentist on-site, while taking the RACF residents to an off-site dental office for examination was the most expensive approach. Conclusions: Real-time teledentistry is feasible and cost-effective, and it is an acceptable alternative to a face-to-face clinical exam for oral health assessment in RACFs. This approach could be used in RACFs where wireless internet connectivity is available.

## 1. Introduction

Older individuals living in residential aged care facilities (RACFs) have some of the poorest levels of oral health in the entire population [1,2,3,4,5], and in Australia this was one of the major findings of a recent Royal Commission into Aged Care [6]. Medically frail older individuals with specialized healthcare needs are a priority group in the Australian National Oral Health Plan 2015–2024 [7]. This study arose from the need to address the unmet oral health needs of residents in Australian RACFs.

Many studies have investigated factors that impact the ability of residents in RACFs to access timely and appropriate oral healthcare [1,2,8,9,10,11,12]. Major barriers include medical co-morbidities, and significant costs for transport to dental clinics for appointments [1,12]. As poor oral health can adversely affect general health [13], close monitoring of oral health in this vulnerable population in aged care is essential. Ideally, the oral health of residents in RACFs would first be assessed on admission to the facility, by trained dental professionals [1,9,14], such as specialists in Special Needs Dentistry (SND), or in some other countries by dentists who have specialized in geriatric dental care [15,16]. Low availability of dentists on-site and a lack of finances for residents both compromise the goals of all residents having an oral health assessment on admission and then again regularly during their time of residence. This challenge has driven the development of oral health assessment tools that can be used by RACF staff, who are not dentally trained [14], such as registered nurses (RNs).

A recent (2021) systematic review examined the range of existing oral health assessment tools used by non-dental professionals in terms of their validity, reliability, feasibility, generalizability, and responsiveness [17]. To ensure a high uptake, the process of completing an oral health assessment needs to be efficient and simple, so that it can be completed by a non-dental health professional in a relatively short period of time [18]. The systematic review concluded that the most reliable tool was the Brief Oral Health Status Exam (BOHSE) [17], and its derivatives, such as the Oral Health Assessment Tool (OHAT) (Appendix A) [14].

The OHAT is used in the Better Oral Health Care Plan (BOHCP), and it is recommended in the Australian National Oral Health Plan 2015–2024 [7,19]. The OHAT is a valid and reliable tool for oral health assessment, in situations where the traditional dental examination is not possible, such as in the setting of RACFs [14]. The OHAT scores the oral soft tissues (lips, tongue, gingivae), the saliva quality and quantity, the status of the teeth (whether natural or artificial teeth), oral cleanliness [14], and also patient self-reported dental pain [20]. For each category, a score of 0 is assigned for health, 1 for changes, and 2 for unhealthy, such that a total score out of 16 is given.

The OHAT can be used to determine when a referral to a dentist is appropriate, and it also informs decisions on the time period between repeated oral health assessments. OHAT scores can be used as a predictor of general health. This was shown in a previous study of mortality risk for geriatric inpatients [21], while another study used OHAT scores as an indicator of oral health and nutritional status [22]. Whilst the OHAT has been implemented in some RACFs, it is unclear whether these assessments are accurate, and whether residents have been referred for and receive the necessary dental care.

A systematic review by Chen et al. explored four distinct models that aim to ensure a sustainable workforce for improving the oral health of residents in RACFs [20]. These models stress the need for a collaborative approach between RACF staff and dental professionals. RACF staff can feel empowered to deliver oral healthcare to residents once they have received sufficient training and assistance from dental professionals.

Previous studies have relied on oral health professionals coming on-site to RACFs to conduct oral health assessments [23,24], or they have involved nursing staff at RACFs undergoing training for onerous amounts of time prior to the study [25]. No previous study has been completed using a model of care based on dental-professional-led training, with clinical support for RNs delivered live during an interactive teledentistry consultation.

Issues around accessing dental care for residents in RACFs can potentially be addressed through teledentistry, by using existing information technology infrastructure, such as wireless internet connectivity. The design of the present study drew on the success of previous successful programs: the 2017 Reach-OHT model [26], the Senior Smiles™ model [27], and a real-time teledentistry project undertaken in rural and regional Australia [13]. Previous studies have utilized trained oral health therapists (OHTs) to manipulate an intra-oral (I/O) camera onsite at the RACF, with images then assessed by a remotely located dentist [13]. Unlike previous studies, the present study used non-dental health professionals in RACFs to operate the I/O camera, under the real-time guidance of a dentist (who was also a specialist in training in SND). This approach connects residents with specialist-level oral healthcare, and it simultaneously trains RNs in how to complete an oral health assessment through real-time teledentistry. This inter-professional relationship between dental and non-dental professionals fosters a collaborative approach that has not been used in previous studies.

Economic considerations for teledentistry (TD) are important for convincing health policymakers that it is a feasible method for assessing the oral health of residents. The costs of TD will vary depending on whether dental clinician input is real-time (synchronous) or delayed (asynchronous) [25]. Real-time input means that feedback can be provided, and it helps build relationships between nursing staff and dental professionals. Previous studies have examined economic factors for TD conducted by RNs, with RNs undergoing training prior to the study [25], or have assessed TD models that use an oral health therapist on-site at the RACF [24]. In contrast, the present study focused on the costs of real-time teledentistry by non-dental health professionals, with both training and assessment occurring simultaneously under the guidance of a dentist.

Based on these considerations, the present study had the following aims:To determine the feasibility of real-time teledentistry conducted by aged care staff under the real-time guidance of a dentist, through a narrative review of the feedback and experiences from participants and from the dentist.To undertake cost estimates of various modes of teledentistry versus on-site clinical examination.

In regard to the context and background for the teledentistry intervention, this study also determined the sociodemographic background, dental status, oral hygiene habits, and dental attendance patterns of the participants. Additionally, for residents only, their medical status and dependence on others for daily oral hygiene care were assessed.

## 2. Materials and Methods

The team consisted of 2 dental specialists (LJW, CLPS), a dental specialist in training (CF), and a senior dentist and researcher (DH), all with more than 10 years of clinical experience. Two public sector RACFs in Brisbane, QLD, Australia were involved in the study—Gannet House and Cooinda House. Both are operated by the Metro North Health and Hospital Service. These facilities care for residents with high care and specialized healthcare needs. As a result, most residents require a substitute decision-maker (33 out of 40 (82.5%) for Gannet House, and 50/60 (83.3%) for Cooinda House).

This study ran from July 2022 to January 2023. Participants included both residents and RACF staff. Participants were recruited by invitations sent with weekly newsletters, and by using emails, flyers, and monthly resident meetings. Potential participants were also identified by speaking to RNs. Once participants were identified and screened for eligibility, formal consent processes followed. The workflow of this study is shown in Figure 1.

A background questionnaire collected the baseline demographic data and clinical characteristics of all participants (Appendix A). All participants underwent a dental assessment using teledentistry, with scoring using the OHAT (Appendix A) [14]. During the teledentistry assessment, feedback regarding the process was collected from participants. To allow for a break between two different assessment methods, a face-to-face (F2F) clinical examination was conducted on-site 4 weeks after the initial teledentistry exam.

The F2F exam was conducted using the Adult Oral Health Assessment form (Appendix A), which was adapted from the WHO Oral Health Assessment [28]. This included scoring oral hygiene based on the presence of plaque deposits using the Sillness and Löe index [29]. The clinical examination scored several items that required the use of dental instruments (and hence, some aspects that were not possible during a teledentistry consultation).

A further OHAT score was allocated by the examining dentist. This was then compared to the OHAT score given at baseline by the RN, and also to the OHAT score from the teledentistry session.

At the end of the study, participants’ feedback was collated through a post-participation semi-structured interview (Appendix A).

All participants received oral health and oral hygiene information during this study. The Oral Hygiene Care Plan (OHCP) (Appendix A) for each participant was updated [9] and oral hygiene information provided [19].

Since the study design placed an emphasis on assessing the feasibility of teledentistry, telehealth facilitators for Queensland Health (QHealth) were consulted in the design phase and when trials of equipment were conducted. The standard telehealth set-up that was used included a QHealth Dell^®^ laptop, using its inbuilt videoconference camera for the F2F components of the consultation. The TD consultation was conducted in real-time, through the online platform Microsoft Teams^®^ version 4.15.58.0. Following equipment trials to optimize image quality for I/O examination during teledentistry, the Mouthwatch I/O camera (Mouthwatch LLC, Metuchen, NJ, USA) was selected. The Mouthwatch I/O camera has a resolution of 640 × 480 pixels, with illumination from 6 inbuilt LED lights, and it is connected to a laptop computer via a USB cable connection [30].

The real-time teledentistry consultation occurred using the laptop computer’s inbuilt camera and then the Mouthwatch I/O camera. Residents and RACF staff (RN, physiotherapist) first met the dental clinician through the normal interface of Microsoft Teams^®^. This ensured that they understood the purpose of the consultation, and it gave those staff the opportunity to ask any questions regarding the set-up of the I/O camera, such as correct placement of the disposable barrier sleeve.

As the teledentistry assessment progressed, staff used the I/O camera under the guidance of the dentist, who provided instructions on how to best position the camera to capture suitable images for each of the various categories in the OHAT (lips, tongue, gums and tissues, saliva, natural teeth, dentures, oral cleanliness, and dental pain). The dentist also asked the staff questions around what they were seeing during the examination (e.g., Is the saliva watery, stringy, frothy, or ropey, or is there no saliva?). The inbuilt LEDs of the I/O camera provided illumination for examination. The I/O camera video feed was viewed by the dentist synchronously as the staff members slowly scanned each area under their guidance. As the staff members were guided to look through each area, any areas of concern that had been raised by the participants were then examined more closely. Overall, the examination process took 20 min per person, including allowing for occasional breaks.

### 2.1. Participant Inclusion/Exclusion Criteria

The inclusion criteria included all consented staff (e.g., RNs, physiotherapists) who cared for residents, and residents in the RACF. All participants were required to complete all components of this study.

To exclude residents with severe cognitive impairment, the cognitive capacity of each resident was rated using the Psychogeriatric Assessment Scale (PAS). PAS scoring was already used in the RACFs, with scores allocated on admission and then updated annually. Any residents with scores ≥ 15 were excluded from this study, as scores of 15 or above indicate severe cognitive impairment [31,32]. Excluding such residents prevented distress to them and ensured those who were included could participate fully.

Staff members from the RACF who were unable to attend all components of this study were excluded.

### 2.2. Setting and Location of Data Collection

After the teledentistry component had been completed online, face-to-face (F2F) clinical examination by a dentist who came on-site followed 4 weeks later.

During TD examinations of RACF residents, the RN who was operating the camera was guided by a dentist (CF) as they captured a video from the I/O camera. For participants who themselves were staff members of the RACF, each of them used the I/O camera in their own mouth, and they were guided by the same dentist.

### 2.3. Interventions

A detailed work instruction manual was developed as a protocol for this study (Appendix A). This addressed the teledentistry procedures and the methods for scoring oral health using the OHAT. RACF staff were already familiar with using online telehealth software. RACF staff followed the work instruction manual when undertaking teledentistry assessments. Troubleshooting of problems occurred in real-time during the telehealth consultation.

Individual advice was given to participants based on the results of the dental assessment. Likewise, advice on oral care needs was tailored to each participant. Oral health and oral hygiene advice included aspects such as the level of support required with oral hygiene and the choice of appropriate oral hygiene procedures and products [33].

### 2.4. Sample Size

The study projection for sample size was based on low expected rates for participation of residents in RACFs [34,35,36], along with recommendations for sample sizes of from 18 to 25 participants [37]. The present investigation was a pilot study to assess feasibility, and it took into account likely attrition rates [38].

### 2.5. Outcomes

The following outcomes were assessed:Overall acceptability of the teledentistry approach from post-participation feedback from all participants: residents and staff at the RACFs and also the dentist. This included the suitability of the oral health assessment instruments used.Estimates of the cost for an oral health assessment delivered through teledentistry versus via an F2F clinical examination by a dentist on-site.Demographic data of the participants, their existing oral hygiene practices, and their past dental attendance pattern. For residents, their medical status and their level of assistance for oral hygiene practices were also noted.

Univariate descriptive analysis was conducted for participant sociodemographic data and the medical status using Jamovi version 2.3 software. Cost estimates were based on current clinical practice scenarios and operational data for public sector health services. Other outcomes, including clinician and participant perspectives on oral health assessments through teledentistry, were descriptive in nature. The quantitative analysis of OHAT score data appears in a subsequent publication in this Special Issue “Validity of real-time teledentistry in residential aged care facilities: An observational feasibility study”.

### 2.6. Ethical Approval

Human ethical approvals for this study were granted from the Royal Brisbane and Women’s Hospital (HREC/2021/QRBW/77399) and from the University of Queensland (2022/HE000091). Approval for participants who required decisions to be made by a substitute decision-maker was obtained from the Queensland Civil and Administrative Tribunal (QCAT) (approval CRL035.21).

## 3. Results

### 3.1. Implementing Teledentistry in RACFs

A streamlined mobile teledentistry set-up was used on a surgical trolley (Figure 2). This included the laptop computer, the I/O camera and its protective barrier sleeve, associated items of protective personal equipment (PPE), and other disposable items required for oral examination in the domiciliary dental care setting [39]. Other than the I/O camera, all other items were readily available at the two RACFs.

### 3.2. Recruitment

As shown in Figure 3, overall, 29 participants were recruited. A total of 24 participants completed this study (16 residents and 8 staff members). Multiple issues were encountered with recruitment. This included the high care needs of residents in this study, which limited eligibility based on the PAS score, since both RACFs had many residents who were ineligible because of cognitive impairment, or because of other impairments that prevented assessment using the PAS score. Some potential participants chose not to participate because of innate negative views towards dental care, whilst others had an end-of-life view and believed that there was no benefit in receiving oral healthcare.

### 3.3. Demographics of Participants

Demographic information for the participants is summarized in Table 1. For residents, there was an equal number of males and females, with an average age of 79.3 years. For RACF staff, there were more female participants, with an average age of 33.3 years.

All RACF staff were brushing their teeth at least twice each day. In contrast, for residents, only 31% brushed at least twice each day. All staff undertook interdental cleaning, whilst only one resident had an interdental cleaning habit. Every participant in this study used a fluoride toothpaste. No participants used dry mouth oral care products.

Barriers to dental care were an issue for 75% of residents, with some citing multiple reasons for inaccessibility (e.g., wheelchair access, hoist transfer, bariatric needs). Other common issues cited by residents were a lack of transportation and dental clinics that were not accessible. Financial difficulties were another major barrier to access (100% staff, 25% residents).

Some 44% of residents had been in the RACF for less than 2 years. Most residents (75%) had more than five chronic medical conditions, and the majority (81%) were taking more than five prescribed medications each day. In line with the eligibility criteria for the study (PAS < 15), 88% of the participants had mild to moderate cognitive impairment.

Almost two-thirds of the participants (63%) experienced no pain or discomfort from their teeth, gingival tissues, or mouth. The majority of the participants (69% of residents and 100% of staff) believed that maintaining good oral health was important.

The level of need for assistance with oral hygiene is summarized in Table 2. This shows information provided by nurses regarding the residents’ needs for assistance with oral hygiene practices, versus the ratings assigned from the dentist. Nurses identified that 44% of residents did not require any assistance with oral hygiene practice, compared to only 25% for the dentist. Overall, the dentist determined that around three-quarters of residents would benefit from some level of assistance, including task breakdown, supervision, or full assistance to carry out oral hygiene practices.

Dental attendance patterns for participants are compared in Table 3. For residents, information was collected on dental attendance patterns before and after entering residential aged care. This was compared to dental attendance for staff. Prior to entering aged care, the dental attendance patterns of residents and staff were similar. Some 63% of residents and 75% of staff attended for preventive dental care. When the dental attendance pattern was dichotomized into regular or irregular attending (attending at least every 2 years) [40], 50% of residents were regular attenders, as were 75% of staff. There was a marked change in patterns of attendance after entering aged care, with only 44% of residents regular attenders, while only 20% of residents still had a regular dentist.

### 3.4. Feasibility of Teledentistry Design

#### 3.4.1. Suitability of Eligibility Criteria

In the original recruitment phase, 50% of participants who consented met the criteria of PAS >15. Residents of both facilities in this study had a high probability of cognitive impairment. In some residents who did not have cognitive impairment, the PAS was unable to be assessed due to sensory or physical impairments.

#### 3.4.2. Suitability of Instruments Used for Dental Assessment

The OHAT was used for dental assessment. OHAT scores were allocated by RNs and recorded in the residents’ medical charts. The teledentistry assessment used information from the real-time video feed. The image quality from the I/O camera was rated as adequate by the dentist who was working remotely. For the F2F clinical examination, the Adult Oral Health Assessment Form was used. This tool gathers more information about the status of the dentition, and it includes a measurement of periodontal status. Such additional information could potentially affect the OHAT score. However, because of the TD arrangements, it was not possible for the RNs to use dental instruments to measure periodontal pocket depths.

### 3.5. Estimated Cost of Teledentistry

The cost breakdown for the current study on conducting oral health assessments through teledentistry versus an on-site F2F clinical examination, and projected costs for scaling up to all 100 residents, is shown in Table 4. The estimated cost for a resident to attend an off-site dental clinic for a clinical examination is also included. The cost reporting framework was adapted from a previous study [24].

When comparing the three scenarios of dental examination, teledentistry was the least costly scenario, as it minimized travel costs for the dental team, residents, and RACF staff. For F2F on-site examinations, as the dental team would be traveling to provide care for several residents at the same facility, this minimized the per person cost of a clinical examination. The off-site clinical examination approach incurred the highest cost for residents.

### 3.6. Oral Health Assessment Through Teledentistry

#### 3.6.1. Clinician Perspectives on the Quality of Images Taken During Teledentistry Assessment

When using teledentistry, the dentist could readily provide instructions to staff through the online platform, to guide them in completing the set-up for the intra-oral camera, and then its operation in the mouth. Overall, the typical time taken for teledentistry assessment, including set-up, pack-up, and oral hygiene instructions, was 20 min. As the assessment occurred live, the dentist could guide the non-dental staff members to ensure that all relevant areas were captured, so that the assessment did not need to be repeated. Videos from the real-time teledentistry assessment were recorded for post-data-collection analysis. Still images were extracted from these video recordings, or screen captures were taken at the time of the consultation. Examples of captured images are shown in Figure 4. The quality was rated as useful for the purpose of an oral health assessment and referral.

#### 3.6.2. RACF Staff Perspective

Staff gave positive feedback regarding teledentistry. They found that it was time-efficient and comprehensive. Staff expressed that logistically, it was far easier to have a potential dental problem assessed through teledentistry, prior to arranging for a resident to travel out to an appropriate community dental clinic or making an arrangement for a domiciliary care visit by a dentist.

During the TD sessions, staff members were actively learning to differentiate between normal and abnormal soft tissues, and they were able to ask questions regarding areas of concern. Through the teledentistry assessment, staff received training, thus helping them recognize normal versus abnormal oral conditions.

The oral health knowledge of staff increased during the consultation, as was evident in their ongoing questions and their improving ability over time to recognize abnormal oral health conditions. The intra-oral camera significantly improved visibility for examining the oral environment. Staff were also shown how to capture still images of abnormal or areas of concern, to form part of documentation of a resident’s oral health conditions.


*“It is really easy to use the camera. The dentist guides you through the entire consultation and tells you exactly what you need to do.”*
(from an RACF staff member)

#### 3.6.3. Residents’ Perspective

Residents found that teledentistry was informative, as they were able to see, from the image on the screen of the laptop computer, inside their own mouth, down to the details of each individual tooth. Positive feedback from the residents was around being able to have a dental consultation within the comfort of their own room, and to have this led by a staff member who they were already familiar with.

One participant had suspected medication-related osteonecrosis of the jaws, which caused much discomfort. Being able to see the bony spicule (Figure 4f) allowed the participant to understand what was happening inside their mouth, and it improved their oral health knowledge.


*“You can see so much inside my mouth on the big screen!”*
(from an RACF resident)

#### 3.6.4. Next of Kin/Family Perspective

Family members also felt positive with regards to the teledentistry consultation. For one resident, prior to her stroke, she was a regular attender at a dentist. Due to limited mobility, the requirements of hoist transfer, and the needs to use a specialized custom-made wheelchair, she would need ambulance transport to attend for a clinical exam. Teledentistry allowed this resident to have an oral health assessment by a dentist, bypassing logistical problems such as her physical impairment or the necessity of ambulance transport.


*“It is really good to be able to have a check-up by the bedside. Without this, my mother would have to go to a dentist on a stretcher with ambulance transport.”*
(from a resident’s family)

### 3.7. Oral Hygiene and Oral Health Advice

Oral hygiene and oral health information could be provided during a teledentistry consultation, as in the setting of a traditional F2F clinical examination. It was natural to discuss oral health conditions as the assessment was occurring, when the participant could see images of their own mouth on the laptop computer screen. Therefore, teledentistry created opportunities to discuss oral hygiene, and oral health information could be delivered at the time of the consultation.

During teledentistry consultations, staff members noticed oral hygiene issues, such as uncleaned dentures and food or plaque that had accumulated on teeth. This actively prompted them to carry out oral hygiene for the residents during teledentistry.

### 3.8. Post-Study Evaluation

Post-study evaluations were conducted for both residents and for RACF staff who were participants. Due to some residents’ high care needs and cognitive impairment, they did not remember the teledentistry consultation when asked about this at the time of their F2F examination some 4 weeks later.

Overall, participants found that having a teledentistry assessment reduced their anxiety about having a dental examination, as it minimized the sights, sounds, smells, and sensations that may trigger dental anxiety during a traditional dental office visit [45]. The small size of the I/O camera (the size of a toothbrush) meant this was considered by them to be a non-invasive device. Participants felt comfortable as the examination occurred at their own pace. Some preferred teledentistry over the F2F exam since they could now see what was happening inside the mouth. Staff gained benefits from the step-by-step prompts provided by the dentist. These prompts ensured that each category of the OHAT was assessed. The real-time feedback given by the dentist provided an opportunity for education and further training of staff members regarding components of oral health assessment.

Several areas of improvement were identified, including providing further guidance on how to best position the I/O camera, especially at the beginning of the teleconsultation. Staff members requested further in-service training around how teledentistry could be implemented in their facilities. They were also interested in how well their existing oral health assessment for residents in the resident’s charts compared to the assessments completed through teledentistry.

## 4. Discussion

The real-time teledentistry set-up used in this study was feasible, and it could readily be implemented as a regular means to assess the oral health of residents in RACFs [24,38]. Clinically, residents in aged care face significant barriers in accessing traditional in-office dental examinations. Teledentistry is a potential solution to resolve this.

All participants in this study, both residents and staff, found TD to be acceptable. The mobile set-up with a laptop computer and an I/O camera could easily be moved to any location in the facility. The real-time advice provided by the remote dentist enhanced the training of nursing staff [24]. It reduced the need for the dentist to be physically present at the RACF [38], yet it still provided residents with access to the dentist’s expertise [46].

In the present study, the remote dentist was a specialist in training in SND. This gave participants access to specialist-level care, bypassing unnecessary travel. It also facilitated appropriate referrals based on the resident’s medical status and their dental needs [39]. Residents completed their dental assessment in the comfort of a familiar environment, with staff who they knew and trusted [1]. These were important benefits, given that many residents find it difficult to leave the RACF due to physical impairments, additional transport requirements, bariatric needs, cognitive impairment, and functional decline.

Real-time images shown on the laptop computer provided all participants with visual information on their own oral health. It increased their awareness, and gave them the opportunity to engage in the process of discerning between normal and abnormal conditions. In this way, it helped to elevate the priority residents gave to seeking timely oral healthcare [47].

Teledentistry could improve time management. Staff working in RACFs often cited inadequate time and training for assessing oral health and providing support for oral hygiene [2,47]. Synchronous teledentistry provided staff with further training, to better support them in assessing oral health using the OHAT [14,19]. With better training, they were more likely to identify oral health conditions that require referral [20]. This ensured more efficient detection of issues by RACF staff [13,38]. The recorded video of the consultation and the associated still image captures documented the resident’s oral health at that one point in time, allowing for better tracking over time [38].

The present study provided insights into time, cost, and convenience aspects of real-time teledentistry consultation for aged care staff in RACF. An oral health assessment is part of routine duties for nursing staff. One of the greatest impacts on time management is when staff or family members need to accompany residents to attend off-site dental examinations at a dental clinic in the nearby community. These visits can also be a major disruption to the resident’s daily routines, especially in the context of dementia [1,24,27,38,48]. General dentists may lack the skills to manage residents with additional and specialized healthcare needs [16]. There was predominantly anecdotal evidence implying that this will then lead to the resident being referred to specialists in SND, without any dental treatment being performed.

Previous studies on the relative costs of teledentistry (asynchronous and synchronous) versus on-site examination suggested that synchronous examination was the costliest method [25]. This is in contrast to the present study, where teledentistry was the most economical modality for assessing oral health. It avoided the need for residents to travel to nearby community clinics, obviated issues with transporting residents to external facilities [1], and eliminated the additional time needed by staff to arrange and attend such appointments with residents [24]. The teledentistry model using specialist-level input also circumvents issues with dentists in general dental practice not having an adequate skillset to manage RACF residents who have additional needs [16].

During teledentistry consultations, issues with video quality (such as dropped frames) could be caused by unstable internet connections [38]. During this study, there was sufficient bandwidth to conduct face-to-face consultation via an online platform. Therefore, potential internet connection and technological problems were not encountered during the present study. To improve the quality of the I/O images captured, future studies can be completed with a 1080-pixel high-definition I/O camera.

In terms of set-up, since the COVID-19 pandemic, most healthcare facilities in developed countries are now equipped with up-to-date technology for telehealth consultations. Teledentistry only requires the additional purchase of an intra-oral camera, which is not costly compared to other models of providing oral healthcare. However, there may be limitations to generalizing this approach for locations without ready access to telehealth technology.

The present results reinforce well-known issues with oral healthcare in elderly institutionalized people, including more irregular patterns of dental attendance and less frequent oral hygiene. Adding to this, aged care facilities may experience lockdowns during pandemics and during outbreaks of influenza or norovirus, further impacting how residents can access timely oral healthcare. Teledentistry conducted by RACF staff working with a remote dentist can provide a “pandemic-proof” approach that ensures that oral health assessments are not neglected for this vulnerable population. Even during a lockdown, oral health assessments could still be undertaken using teledentistry, with RACF staff operating the I/O camera. The information gained from the assessment can also inform preventive oral health and oral hygiene advice that the resident will receive.

Regarding the perceived importance of oral health, nearly one-third of residents had neutral views on this or considered that it was unimportant. When discussing this issue with residents, we found that many had adopted an end-of-life view towards their time in the RACF. Residents are often medically frail and follow a slow dwindling trajectory of decline with minimal functional reserve. Under such conditions, an acute condition linked to poor oral health, such as aspiration pneumonia, may prove fatal [49]. Therefore, it is suggested that older adults should access dental services early in this stage of life to address their specific dental needs (and needs for prevention) before they transition to an RACF.

In an RACF, it is important to tailor oral health education to residents, and to focus on the benefits of preventive oral healthcare, including how this can minimize the systemic impacts of poor oral health [50,51,52]. Input from specialists in SND can be very important since they are trained in the provision of rational dental care in the context of residents with additional and specialized healthcare needs, including in palliative care settings.

Dental attendance patterns for residents often change upon entry to an RACF, and this was seen in the present study, with many residents losing contact with their regular dentist after entering the RACF, and their attendance pattern changing from regular to irregular [40]. Some 50% of the residents in the current study had not seen a dentist since entering the RACF. The reasons behind this may include dentists not having the skillset to provide care for residents with additional healthcare needs [2,11,46], problems with transport to dental appointments, and dental clinics not being readily accessible. These issues have also been noted in previous studies [2,10,11,12].

In the present study, most residents were taking five or more prescribed medications, and this polypharmacy has implications for salivary gland hypofunction [53]. Both the teledentistry assessment and the F2F clinical examination found a reduction in saliva production at rest, and oral mucosal changes caused from long-standing oral dryness, in line with what has been reported in the literature [1,53,54,55,56,57]. In contrast, the RNs performing the OHAT assessment rated all residents as having normal saliva flow, with minimal changes to the oral mucosa. As a result, the oral care plans prepared by the RNs had no recommendations for using dry mouth products or saliva substitutes. This failure to identify reductions in saliva production is concerning. Saliva has multiple protective roles, and reduced production at rest has major implications, such as a greater dental caries risk [53] and a worsening of oral health-related quality of life [58]. It is therefore important to educate RACF staff on how to assess salivary characteristics, so they can better identify and manage issues relating to salivary gland hypofunction.

Another key area was the need for assistance with oral hygiene care. The type of assistance needed can range from reminders and prompts, through to supervision, and then all the way to full assistance where RACF staff undertake the oral hygiene procedures for the resident [19]. Teledentistry is useful for showing RACF staff where oral hygiene procedures are not being carried out adequately. In the present study, teledentistry provided the opportunity to update the oral hygiene care plan for each resident. As staff were examining the resident’s mouth, they noticed oral hygiene issues that required further attention. This encouraged them to make improvements to oral hygiene practices for residents (e.g., cleaning a denture, charging an electric toothbrush) and to update oral hygiene plans. The present study differs from previous studies in that the approach used directly empowered RACF staff to increase their confidence when examining the oral cavity [10].

The present study is the first where synchronous teledentistry has been conducted by non-dental health professionals under the guidance of a dentist. It is feasible to readily apply this in an RACF setting, to simultaneously train and assess oral health in residents in RACFs. The results from this pilot study can serve as a foundation for larger projects.

There were several limitations in the present study, with the main one being the sample size. RACFs are a challenging setting in which to conduct research [38,59]. To improve participation rates, dedicated research personnel located on-site can act as local champions of oral health, to boost participation [27,60]. These could be OHTs, as their skill set includes health promotion [61,62], and their effectiveness in the RACF setting has been shown [23].

Another limitation was that of potential bias. This could be reduced if different examiners completed the teledentistry assessment and the face-to-face examination. In the TD consultation, the use of probes and other dental instruments was not possible, nor were additional diagnostic tools such as X-rays available. Hence, oral health assessments cannot completely replace comprehensive clinical examinations; however, they can serve as a referral pathway, to connect residents in RACFs who have limited access to oral healthcare to professional-level oral health advice.

## 5. Conclusions

This pilot study demonstrates that mobile teledentistry is feasible and cost-effective for assessing the oral health of residents of RACFs. Teledentistry addresses many of the barriers for continuity of oral health assessment in this population. It also improves the oral health literacy of both staff and residents. Overall, in this case, the teledentistry set-up with real-time consultation using an intra-oral camera operated by RACF staff under the guidance of a qualified dentist was effective for oral health assessment and education.

## Figures and Tables

**Figure 1 healthcare-12-02216-f001:**
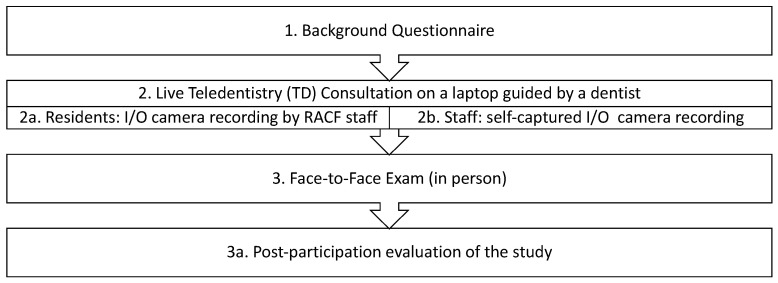
Summary steps of the project.

**Figure 2 healthcare-12-02216-f002:**
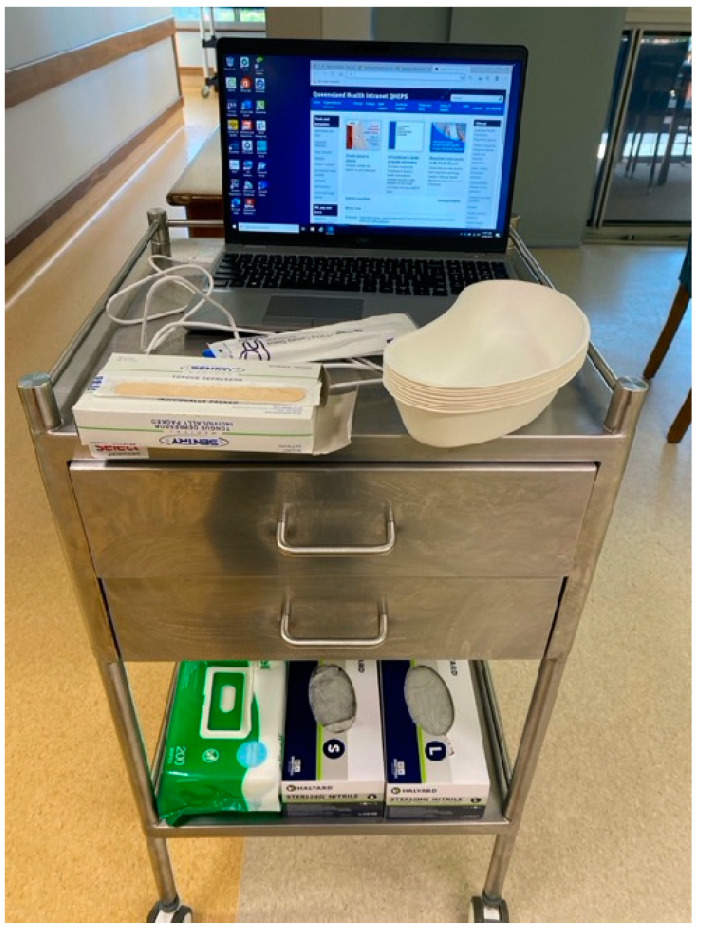
Mobile teledentistry set-up that was brought to each resident’s room. Essential equipment includes laptop, I/O camera (+protective barrier), gloves, disinfectant wipes, disposable trays, and tongue depressors.

**Figure 3 healthcare-12-02216-f003:**
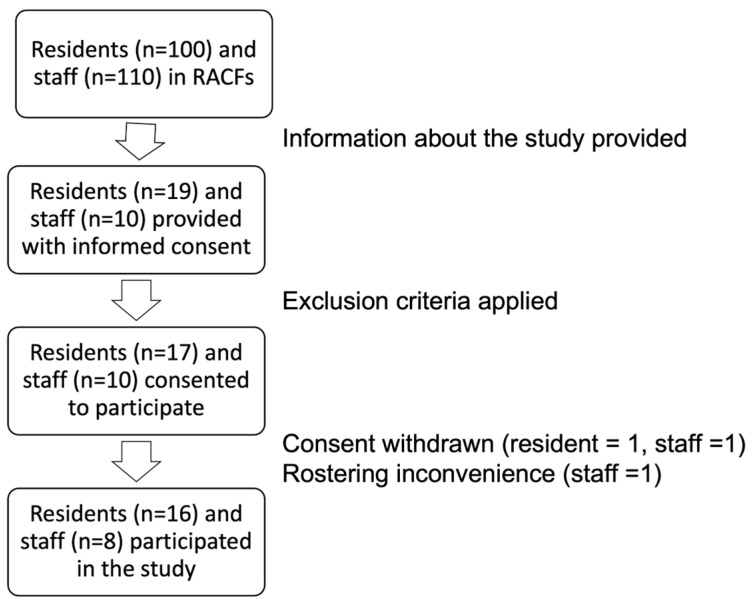
Flow diagram showing participants from the RACFs in this study.

**Figure 4 healthcare-12-02216-f004:**
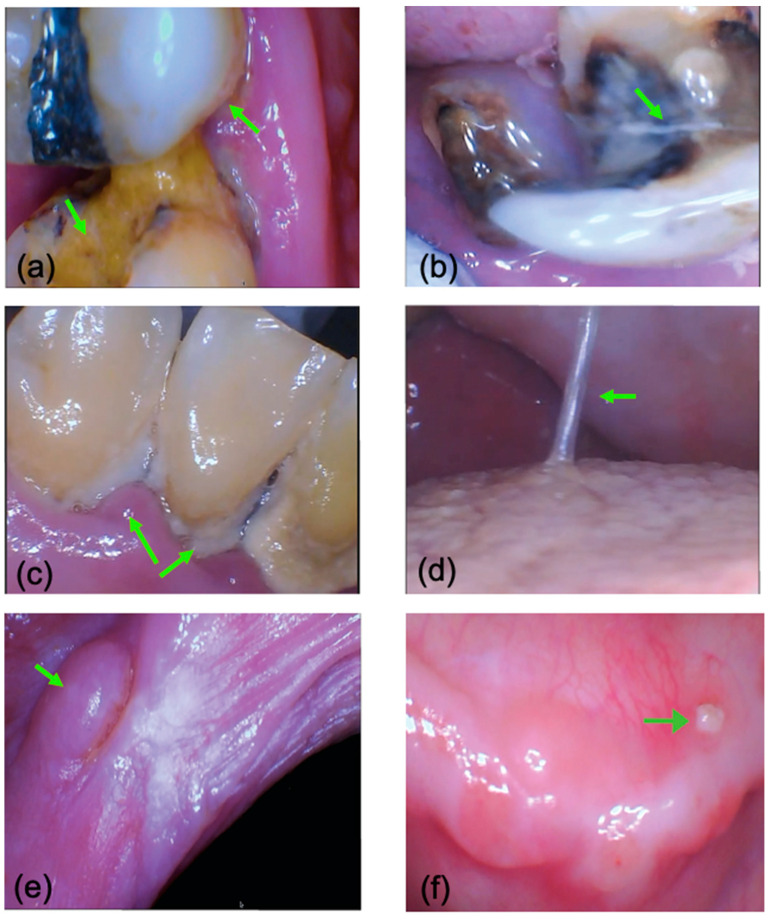
Images captured from the I/O camera during a teledentistry consultation. Images were video frames or screen captures. (**a**) carious lesions including root caries in teeth in the upper right quadrant; (**b**) a retained root and a crack in a tooth in the lower left quadrant; (**c**) poor oral hygiene, with abundant calculus build-up and poor periodontal health in lower anterior teeth; (**d**) stringy viscous saliva on the tongue; (**e**) soft tissue lesions on the left buccal mucosa; (**f**) an area suspected of being medication-related osteonecrosis of the jaws on the lower left edentulous ridge.

**Table 1 healthcare-12-02216-t001:** Demographics of participants in this study, medical statuses of residents, and oral hygiene practices of participants.

	Residents		Staff	
	N	%	N	%
Demographics				
Age range				
Range (min–max)	61–93		24–43	
Mean ± SD	79.3 ± 8.68		33.3 ± 6.16	
Gender				
Male	8	50	3	37.5
Female	8	50	5	62.5
Medical status				
Time since admission				
0–24 months	7	43.8		
25–60 months	6	37.5		
>60 months	3	18.8		
No. of chronic medical conditions				
3–4	4	25		
5+	12	75		
No. of medications				
1–4	3	18.8		
5–8	5	31.2		
9+	8	50		
PAS score				
Minimal (0–3)	6	37.5		
Mild (4–9)	8	50		
Moderate (10–15)	2	12.5		
Oral hygiene practices				
Frequency of brushing				
At least 2 times per day	5	31.2	8	100
Some brushing habit	7	43.8	0	0
Never cleaned	4	25	0	0
Toothpaste used				
Toothpaste 1000 ppm fluoride	11	68.8	8	100
Toothpaste >1000 ppm fluoride	1	6.3	0	0
Mouthwash	1	6.3	3	37.5
Oral hygiene aids				
Manual toothbrush	11	68.8	8	100
Electric toothbrush	1	6.3	0	0
Interdental brush/floss/fossettes	1	6.3	8	100
Other aspects of oral health				
History of dental pain/discomfort				
Yes	6	37.5	3	37.5
No	10	62.5	5	62.5
Importance of oral health				
Important	11	68.8	8	100
Not important	5	31.3	0	0
Barriers to accessing dental care				
Yes	12	75	3	37.5
No	4	25	5	62.5
Reason for barriers				
Dental providers not available	7	43.8	0	0
Financial difficulties	3	25	3	100
No transport to dental appointments	5	41.7	0	0

**Table 2 healthcare-12-02216-t002:** Level of assistance with oral hygiene practices, reported in assessments by RNs and by a dentist.

	RN OHCP		Dentist OHCP	
Level of Assistance	N	%	N	%
No assistance needed	7	43.8	4	25
Reminding, prompting, task breakdown, set up of TB/TP	4	25	7	43.8
Supervision, checking	3	18.8	2	12.5
Full assistance	2	12.5	3	18.8

RN = registered nurse, TB = toothbrush, TP = toothpaste, OHCP = oral hygiene care plan.

**Table 3 healthcare-12-02216-t003:** Dental attendance patterns of participants.

	Residents		Staff	
	N	%	N	%
Visiting pattern BEFORE entering RACF				
Regular pattern	8	50		
Irregular pattern	8	50		
Regular dentist				
Yes	8	50		
No	8	50		
Reason for dental visit				
Dental exam, check-up	10	62.5		
Emergency/other	6	37.5		
Current visiting pattern, i.e., AFTER entering RACF for residents				
Regular pattern	7	43.8	6	75
Irregular pattern	9	56.2	2	25
Regular dentist				
Yes	3	18.8	3	37.5
No	13	81.2	5	62.5
Reason for dental visit				
Dental exam, check-up	5	31.3	6	75
Emergency/other	11	68.7	2	25

**Table 4 healthcare-12-02216-t004:** Summary of costs associated with teledentistry, face-to-face domiciliary exam (F2F exam), and clinical examination (i.e., attending an appointment with a dentist at a dental clinic) for a resident.

	Teledentistry		F2F Exam on Site		Clinical Exam off Site	
Equipment						
Mouthwatch I/O camera USD 299	I/O camera	447				
Camera sleeves USD 29/100 sleeves	1 sleeve	0.45				
Single-use exam pack AUD 68/25 kits			Exam pack	2.72	Exam pack	2.72
Total/resident		35.1		2.72		2.72
RACF cost						
RN5.1 AUD 40.52/h	20 min	13.5	20 min	13.5	2 h	81
Taxi min return					Taxi	50
Total RACF cost per resident		13.5		13.5		131
Dental travel cost						
DO2.1 AUD 69.72/h; CA1.1 (DA) AUD 21.2/h			1.5 h × (DO + DA)	136		
Travel to facility/day			916 (ADA code)	76		
Total cost for four exams/day/resident				53		
Projected for eight exams/day/resident				26.5		
Total cost 16 residents	Teledentistry	670	F2F Exam	1100	Clinical Exam	2140
Projected cost 100 residents	Teledentistry	1800	F2F Exam	4300	Clinical Exam	13400

Costs are shown in AUD unless stated otherwise (AUD 1 = USD 0.66). For TD, the initial outlay cost was purchasing the I/O camera for TD. For F2F on-site examinations or clinical examinations off site, there were ongoing costs for dentists to deliver domiciliary care, or ongoing costs to the RACF for bringing residents to community dental clinics. Staff costs were based on QHealth awards (as of April 2023): for registered nurses, a nurse Grade 5 Level 1 (RN5.1) at AUD 40.52/h [41], for a dentist, senior dentist level 1 (DO2.1) at AUD 69.72/h [42], and a dental assistant at Clinical Assistant Level 1(CA1.1) at AUD 21.2/h [43]. The cost to the dental service was calculated based on dentist and dental assistant pro rata wages for travel time to the facility, and the Australian Dental Association (ADA) item code of 916 for travel to provide services, using the fees set out by the Department of Veterans’ Affairs [44].

## Data Availability

Summary data are available upon reasonable request to the corresponding author.

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
