# Peer review of "A Pilot Study to Assess the Feasibility of Real-Time Teledentistry in Residential Aged Care Facilities"

_healthcare, 2024, doi:10.3390/healthcare12222216_

Round 1
Reviewer 1 Report
Comments and Suggestions for Authors
Real-time teledentistry in Residential aged care facilities: A feasibility study
Reviewer Report
The authors of this study investigated the feasibility of assessing the oral health of elderly care home residents using teledentistry, with an intra-oral camera and the Oral Health Assessment Tool (OHAT). They reported that oral health assessments of these elderly patients via teleconferencing were feasible and cost-effective. In this respect, the study provides important insights into the evaluation of oral health in a significant patient group. However, there are aspects of the study that require improvement from both a scientific and formal perspective. My detailed opinions are as follows:
Abstract
- Line 8-10: It was stated as Background but it was not visible. The aim of the study was stated directly. If this was the case, the Aim subheading should be written directly instead of the Background subheading. Or the Background should be stated in one sentence and then the aim should be stated. And the subheading should be updated as Background and Aim.
- The abstract should be more detailed, for example, the number of participants, their distribution by gender, mean age, etc.
- Was the real-time teledentistry examination performed by telephone or on a computer screen? Some detail should also be given regarding the method.
- Title and Abstract should be consistent. Was the subject solely based on cost calculations? What was determined by the examination with this method? Was an effective treatment option recommendation provided? What were the areas, teeth, etc. that could not be examined? Full understanding should be ensured.
Introduction:
- Very poorly documented. More information should be provided.
- Why was this study needed? What gap in the literature was it intended to fill? Should be highlighted.
- Line 55 ‘1.1 Study design considerations’: This subheading content is not suitable to be presented under the Introduction section. Please delete and merge. Keep the background statements in the Introduction section, as stated under this subheading. And improve the Discussion by moving the statements related to the literature to the Discussion section.
- Provide more information about OHAT.
Materials and Methods:
- Inclusion and exclusion criteria should be clearly stated.
- Ethical approval should also be stated here.
- Since the screen image should be obtained in high resolution during the teleconference for the examination, the technical specifications of the camera should be detailed a little more.
- The experience of the examiners should be stated.
- It should be clearly stated how descriptive and statistical results were presented.
Results:
- There are spelling and grammar errors. They should be carefully corrected and revised according to the journal rules (especially the writing of references, throughout the manuscript).
- The images captured in Figure 4 should be revised to be a little further away and clearer. It was not fully understood which region they were from.
- Figure 4 should be presented after the place it is mentioned in the text.
- The space after line 316 on page 10 should be filled by moving the text continuing on the previous page. The manuscript should be revised according to a systematic order.
Discussion:
- The limitations and strengths of the study should be stated.
- In particular, the inability to perform additional diagnostic tools (radiographs, tomography, clinical examination, etc.) required for comprehensive examination in teledentistry should be noted.
- Please merge subheading 4.1 with the Discussion.
Conclusion:
- This section is sufficient.
References:
- Review references 25, 38, 39, 40, 41, and 42 according to journal rules.

Moderate editing of English language required.
Author Response
comments (in italics) and response
NB additions/ amendments have been highlighted in the manuscript
Abstract
Line 8-10: It was stated as Background but it was not visible. The aim of the study was stated directly. If this was the case, the Aim subheading should be written directly instead of the Background subheading. Or the Background should be stated in one sentence and then the aim should be stated. And the subheading should be updated as Background and Aim.
changed to background and aim. The abstract has been edited
- The abstract should be more detailed, for example, the number of participants, their distribution by gender, mean age, etc.
This has been added where possible, however taking account word limits
- Was the real-time teledentistry examination performed by telephone or on a computer screen? Some detail should also be given regarding the method.
This is discussed in p4 of 19 line 137 - 146. The real-time teledentistry examination was performed on a laptop online via Microsoft Teams. The Mouthwatch I/O camera specs have been added. Figure 1 has been updated to make this more clear
- Title and Abstract should be consistent. Was the subject solely based on cost calculations? What was determined by the examination with this method? Was an effective treatment option recommendation provided?
Outlined in abstract p1 line 9 - 14. There were 2 components:
- The feasibility of setting up real-time teledentistry as an alternative means of traditional in-office face-to-face examination
- Cost estimates to deliver oral health assessment 1. Teledentistry 2. Face-to-face examination 3. Domiciliary care were calculated
What were the areas, teeth, etc. that could not be examined? Full understanding should be ensured.
P1 line 12 - the assessment was recorded using the Oral Health Assessment Tool via the 8 categories: 1. Lips 2. Tongue 3. Gums and tissues 4. Saliva 5. Natural Teeth 6. Dentures 7. Oral cleanliness 8. Dental pain
Introduction:
- Very poorly documented. More information should be provided.
Introduction edited to address the follow and addition based on the next point as well:
- The necessity of oral health assessment in the target population i.e., residents in RACF
- Reason for needing tools for oral health assessment away from traditional means of dental examination
- Models of care to address the oral health needs of residents in RACF
- The necessity of economic considerations and that previous studies are outdated
Please also see addition based on the points below to address the gaps this study filled.
- Why was this study needed? What gap in the literature was it intended to fill? Should be highlighted.
Addition of how this study addressed the gaps of the literature. i.e. how this is novel in providing real-time teledentistry consultation with a training dental specialist, and providing staff training in oral health assessment simultaneously
Explained the different TD models used, and previous work, and how this study is different
- Line 55 ‘1.1 Study design considerations’: This subheading content is not suitable to be presented under the Introduction section. Please delete and merge. Keep the background statements in the Introduction section, as stated under this subheading.
Merged.
And improve the Discussion by moving the statements related to the literature to the Discussion section.
moved
- Provide more information about OHAT.
Additional information and references added line 63 - 70
Materials and Methods:
- Inclusion and exclusion criteria should be clearly stated.
Added in the text line 175 - 185
- Ethical approval should also be stated here.
Added in the text line 226 - 230
- Since the screen image should be obtained in high resolution during the teleconference for the examination, the technical specifications of the camera should be detailed a little more.
the pixel resolution of the camera has been added in line 153-156
- The experience of the examiners should be stated.
Clinical experience of author have been added line 112 - 114
- It should be clearly stated how descriptive and statistical results were presented.
added details in line 219 - 225
Results:
- There are spelling and grammar errors. They should be carefully corrected and revised according to the journal rules (especially the writing of references, throughout the manuscript).
This has been updated and amended.
Correct referencing has been amended now to include reference prior to punctuation.
MDPI style and Endnote style has been modified
- The images captured in Figure 4 should be revised to be a little further away and clearer. It was not fully understood which region they were from.
Please understand the images are screen captures or images from video frames. These are not still images, therefore, it is not possible to present images a little further away. To make this clearer, the caption has been revised so that the region of the mouth the images came from are described. Due to the size of the intra-oral camera, and to ensure that the photographs meet minimal publication standards, this more zoom in view has been chosen to ensure that clinically acceptable and useful images are presented
- Figure 4 should be presented after the place it is mentioned in the text.
This has been amended
- The space after line 316 on page 10 should be filled by moving the text continuing on the previous page. The manuscript should be revised according to a systematic order.
See point above and amendment has been made
Discussion:
- The limitations and strengths of the study should be stated.
Added in discussion line 547-550 and 557-558
Limited sample size, need to test same concept at a larger scale, the use of high resolution 1080p HD I/O cameras for higher resolution, potential directions for future work
- In particular, the inability to perform additional diagnostic tools (radiographs, tomography, clinical examination, etc.) required for comprehensive examination in teledentistry should be noted.
Emphasized that this is an assessment and it is a basis for a referral, acknowledged this as a limitation
- Please merge subheading 4.1 with the Discussion.
merged
References:
- Review references 25, 38, 39, 40, 41, and 42 according to journal rules.
Completed to MDPI referencing style
Reviewer 2 Report
Comments and Suggestions for Authors
First of all, I would like to greet the authors and congratulate them on the theme and work done. The study appears correctly performed and written without logical or factual errors.
Authors have well revised several issues; I think the article is very well written with an extensive and updated bibliography.
The following comments are addressed and require minor modifications to enhance the quality of the manuscript:
I suggest the authors replace, in title, the word Feasibility for Cohort study.
The introduction is complete and elucidative.
-Does the study fill any gaps in this field? What is new compared with previous studies?
The methodology is clear and without apparent errors.
-How did you calculate sample size?
The results are correctly presented, being objective but exhaustive and well perceptible through the tables presented.
The discussion confronts your results with the existing bibliography and the conclusions are in accordance with the results and respond to the proposed objectives.
-What were the major limitations of your study?
Author Response
comments (in italics) and response
NB additions/ amendments have been highlighted in the manuscript
I suggest the authors replace, in title, the word Feasibility for Cohort study.
Title has been modified and replaced with Cohort study
-Does the study fill any gaps in this field? What is new compared with previous studies?
Reinforce that the TD model used live TD with remote dentist watching and giving feedback and RN on site operating the I/O camera, this is different from past studies
This has been added throughout the introduction
-How did you calculate sample size?
This is a pilot study, therefore we do not need to test any hypothesis, therefore calculating a sample size is unnecessary
-What were the major limitations of your study?
Added in discussion line 550 - 562
limited sample size, need to test same concept at a larger scale, using high resolution 1080p HD I/O cameras, this is an assessment only so traditional means of oral examination with a probe or diagnostic tests such as radiographs are
Reviewer 3 Report
Comments and Suggestions for Authors
Dear authors
Thank you for your hard work, doing solid research work is not an easy task, and you have given your time and resources to complete this work.
I have found some points regarding your paper:
1. Can you please improve clarity on how feasibility was specifically measured beyond technical effectiveness?
2. Can you include additional information about potential recruitment challenges and how they were addressed. Any bias?
3. Provide more details on how randomization was ensured (e.g., any biases) to improve transparency.
4. Is it possible to provide a power analysis or a more explicit justification for this sample size? It could make the study more robust.
5. Provide more quantitative measures of feasibility beyond participant feedback (such as, time efficiency, success rates for capturing accurate diagnostic images).
6. Expand the limitations, there are more possible limitations such as the potential for technical difficulties (internet connection, video quality) and the limited generalizability to other types of RACFs with different populations.
7. It is just a suggestion, is it possible to discuss the possibility of use of AI to improve the quality of teledentistry for patients and staffs?
Author Response
comments (in italics) and response
NB additions/ amendments have been highlighted in the manuscript
- Can you please improve clarity on how feasibility was specifically measured beyond technical effectiveness?
comment on acceptance by staff and by patients as well as by the dentist doing the TD component remotely, emphasize access, lowered costs - added to line 443 - 437, 441 - 444; 482 - 490
- Can you include additional information about potential recruitment challenges and how they were addressed. Any bias?
comment who what types of patients agreed and were these representative of RACF residents (age, gender, degree of medical complexity or impairment) Added in line 247.- 250
bias added into limitations at the end of the discussion
- Provide more details on how randomization was ensured (e.g., any biases) to improve transparency.
This was removed, and explained more in line 399 - 406.
- Is it possible to provide a power analysis or a more explicit justification for this sample size? It could make the study more robust.
This is a pilot study, therefore we do not need to test any hypothesis, therefore calculating a sample size is unnecessary
- Provide more quantitative measures of feasibility beyond participant feedback (such as, time efficiency, success rates for capturing accurate diagnostic images).
comment on time taken - line 346
images were clinically useful for an assessment. Please see images in Fig 4, it is useful for the purpose of an assessment and referral. Line 352
TD did not need to be repeated due to live feedback. Line 348
- Expand the limitations, there are more possible limitations such as the potential for technical difficulties (internet connection, video quality) and the limited generalizability to other types of RACFs with different populations.
camera quality influences image resolution - line 477
local wifi network influences bandwidth; local NBN connection influenced bandwidth. there was enough bandwidth to talk face to face on a videoconference using Teams then there was enough to do TD - please see line 474 - 475
generalizability: this pilot study can be a foundation to complete larger project. it can be generalisable
- It is just a suggestion, is it possible to discuss the possibility of use of AI to improve the quality of teledentistry for patients and staffs?
AI could help process stored still images for key features and stored video to recognise features and even better AI could be to compare images or videos from different time points - it is critical though to get comparable views which may be hard for non expert users of I/O cameras
Round 2
Reviewer 1 Report
Comments and Suggestions for Authors
Thank you to the authors for their revisions. They have sufficiently addressed all the revisions.
Comments on the Quality of English LanguageMinor editing of English language required.
Author Response
Please change the title to "A pilot study to assess the feasibility of real-time teledentistry in residential aged care facilities".
This has been changed
Please revise the conclusion and make it short because this is a pilot study to show the feasibility of real-time teledentistry in a small group of participants.
Conclusion has been shortened
Minor editing of English language required
The entire document has been reviewed